# All-trans retinoic acid induces synaptic plasticity in human cortical neurons

**Maximilian Lenz[1], Pia Kruse[1], Amelie Eichler[1], Jakob Straehle[2], Jürgen Beck[2,3], Thomas Deller[4], Andreas Vlachos[1,3,5]\***

[1]Department of Neuroanatomy, Institute of Anatomy and Cell Biology, Faculty of Medicine, University of Freiburg, Freiburg im Breisgau, Germany; [2]Department of Neurosurgery, Medical Center and Faculty of Medicine, University of Freiburg, Freiburg im Breisgau, Germany; [3]Center for Basics in Neuromodulation (NeuroModulBasics), Faculty of Medicine, University of Freiburg, Freiburg im Breisgau, Germany; [4]Institute of Clinical Neuroanatomy, Dr. Senckenberg Anatomy, Neuroscience Center, Goethe-University Frankfurt, Freiburg im Breisgau, Germany; [5]Center Brain Links Brain Tools, University of Freiburg, Freiburg im Breisgau, Germany

**Abstract** A defining feature of the brain is the ability of its synaptic contacts to adapt structurally and functionally in an experience-dependent manner. In the human cortex, however, direct experimental evidence for coordinated structural and functional synaptic adaptation is currently lacking. Here, we probed synaptic plasticity in human cortical slices using the vitamin A derivative all-trans retinoic acid (atRA), a putative treatment for neuropsychiatric disorders such as Alzheimer's disease. Our experiments demonstrated that the excitatory synapses of superficial (layer 2/3) pyramidal neurons underwent coordinated structural and functional changes in the presence of atRA. These synaptic adaptations were accompanied by ultrastructural remodeling of the calcium-storing spine apparatus organelle and required mRNA translation. It was not observed in synaptopodin-deficient mice, which lack spine apparatus organelles. We conclude that atRA is a potent mediator of synaptic plasticity in the adult human cortex.

**\*For correspondence:**
andreas.vlachos@anat.uni-freiburg.de

## Introduction

The ability of neurons to express plasticity by responding to specific stimuli with structural and functional changes is critical for physiological brain function (*Citri and Malenka, 2008*). Over the past few decades, cellular and molecular mechanisms of synaptic plasticity have been extensively studied across various animal models (*Ho et al., 2011*). However, direct experimental evidence for coordinated structural and functional synaptic changes in the adult human cortex is lacking (*Mansvelder et al., 2019*; *Verhoog et al., 2016*). It thus remains unclear whether the structural and functional properties of human cortical neurons adapt similarly to those in the rodent brain.

Vitamin A (all-trans retinol) and its metabolites have recently been linked to physiological brain functions such as axonal sprouting, synaptic plasticity, and modulation of cortical activity (*Drager, 2006*; *Shearer et al., 2012*). Specifically, all-trans retinoic acid (atRA), which is used clinically in dermatology and oncology (*Dobrotkova et al., 2018*; *Hu et al., 2009*), has been studied for its neuroprotective and plasticity-promoting effects in animal models (*Chen et al., 2014*; *Koryakina et al., 2009*). Recent studies have evaluated the effects of atRA in patients with brain disorders associated with cognitive dysfunction, including Alzheimer's disease, Fragile X syndrome, and depression (*Bremner et al., 2012*; *Ding et al., 2008*; *Zhang et al., 2018*). For example, alterations in retinoic acid-mediated synaptic plasticity have been reported in neurons derived from inducible

**eLife digest** The brain has an enormous capacity to adapt to its environment. This ability to continuously learn and form new memories thanks to its malleability, is known as brain plasticity. One of the most important mechanisms behind brain plasticity is the change in both the structure and function of synapses, the points of contact between neurons where communication happens. These sites of synaptic contact occur through microscopic protrusions on the branches of neurons, called dendritic spines. Dendritic spines are very dynamic, changing their shape and size in response to stimuli.

Previous studies have shown that alterations in synaptic plasticity occur in various animal models of brain diseases. However, it remains unclear whether human cortical neurons express synaptic plasticity similarly to those in the rodent brain. Recently, a derivative of vitamin A has been linked to synaptic plasticity. In addition, several studies have evaluated the effects of this derivative in patients with cognitive dysfunctions, including Alzheimer's disease, Fragile X syndrome, and depression. However, there is no direct experimental evidence for synaptic plasticity in the adult human cerebral cortex related to vitamin A signaling and metabolism.

To investigate this, Lenz et al. used human cortical slices prepared from neurosurgical resections and treated them with a solution of the vitamin A derivative all-trans retinoic acid for 6-10 hours. Lenz et al. employed a variety of techniques, including patch-clamp recordings to measure neuron function as well as different types of microscopy to evaluate structural changes in dendritic spines. These experiments demonstrated that the derivative promoted the synaptic plasticity in the adult human cortex. Specifically, it increased the size of the dendritic spines and strengthened their ability to transmit signals. In addition, Lenz et al. found that the spine apparatus organelle – a structure found in some dendritic spines – was a target of the vitamin A derivative and promoted synaptic plasticity.

These findings advance the understanding of the pathways through which vitamin A derivatives affect synaptic plasticity, which may aide the development of new therapeutic strategies for brain diseases. More generally, the results contribute to the identification of key mechanisms of synaptic plasticity in the adult human brain.

pluripotent stem cells (iPSCs) generated from Fragile X syndrome patients (*Zhang et al., 2018*). However, direct experimental evidence for atRA-mediated effects on synaptic plasticity of principal neurons in the adult human cortex is lacking.

Here, we used human cortical slices prepared from neurosurgical resections to assess atRA-mediated changes in the structural and functional synaptic properties of layer 2/3 pyramidal neurons. In this context, we also evaluated the role of the actin-modulating protein synaptopodin (*Mundel et al., 1997*), an essential component of the spine apparatus organelle (*Deller et al., 2003*), which is a key regulator of synaptic plasticity in the rodent brain (*Deller et al., 2003*; *Segal et al., 2010*; *Vlachos et al., 2013*) and has recently been linked to the cognitive trajectory in human aging (*Wingo et al., 2019*).

## Results

### All-trans retinoic acid treatment of human cortical slices

Cortical access tissue samples from eight individuals who underwent clinically indicated neurosurgical procedures, such as for tumors or epilepsy, were experimentally assessed in this study (details provided in *Supplementary file 1*). Acute cortical slices were treated for 6–10 hr with atRA (1 µM) or vehicle-only, and superficial (layer 2/3) pyramidal neurons were recorded in a whole-cell configuration (*Figure 1A,B*). While no significant differences in active or passive membrane properties were detected between the two groups (*Figure 1C–E*), a robust increase in the amplitudes of glutamate (i.e., α-amino-3-hydroxy-5-methyl-4-isoxazolepropionic acid, AMPA) receptor-mediated spontaneous excitatory postsynaptic currents (sEPSCs) was observed in the atRA-treated slices (*Figure 1F,G*; see also *Figure 1—figure supplement 1*). The mean sEPSC frequency was not significantly different between the two groups (*Figure 1G*). These results demonstrate an atRA-mediated strengthening

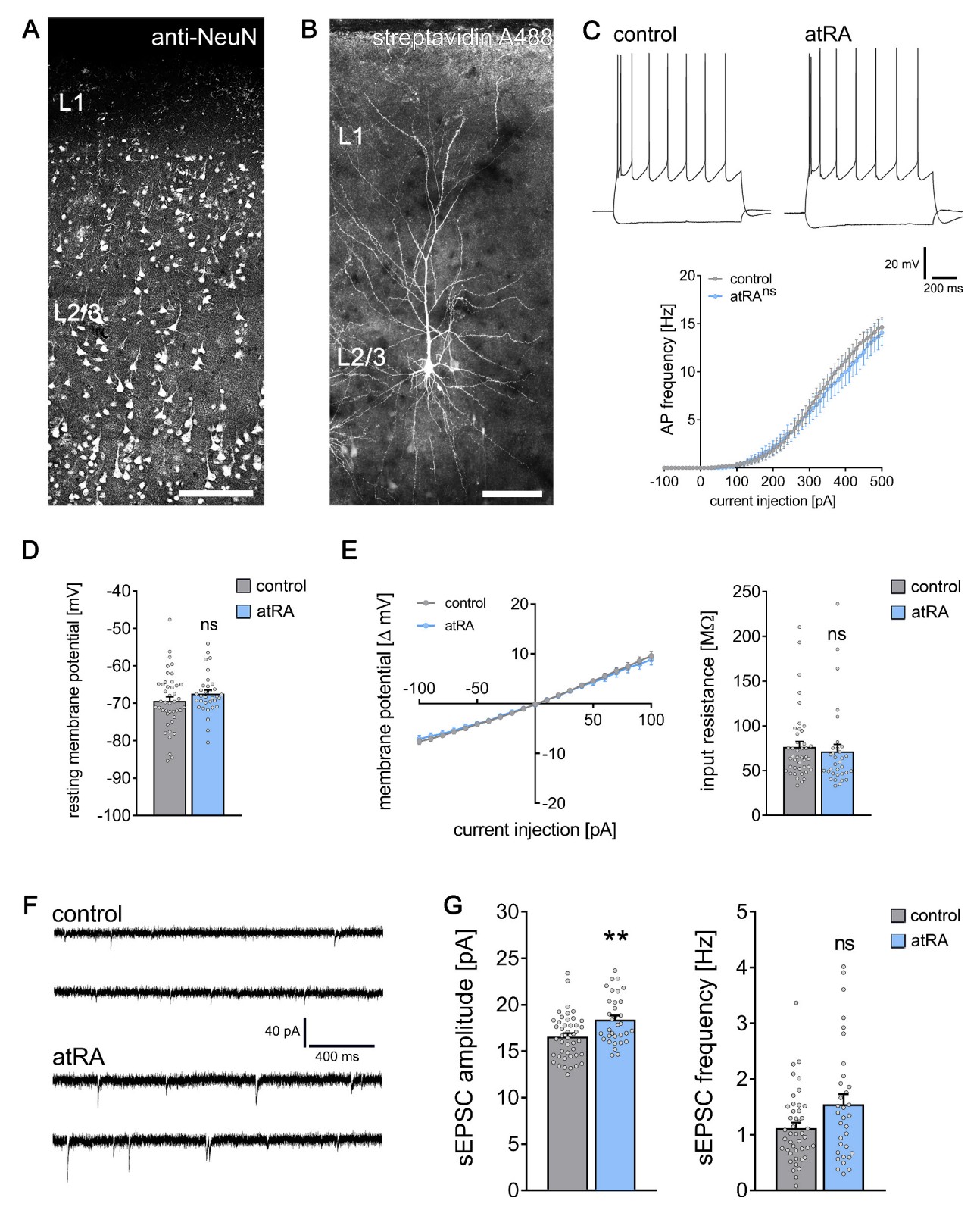

**Figure 1.** All-trans retinoic acid (atRA) induces plasticity of excitatory synapses in human cortical slices. (**A**) A representative human cortical slice stained for NeuN. Scale bar = 200 μm. (**B**) Recorded and post hoc-labeled superficial (layer 2/3) pyramidal neuron. Scale bar = 100 μm. (**C**) Sample traces of input/output-curves of cortical neurons from atRA- (1 μM, 6–10 hr) and vehicle-only-treated slices (responses to −100 pA and +350 pA current injection illustrated). Action potential frequency of human neocortical neurons from atRA- and vehicle-only-treated slices ($n_{control}$ = 38 cells, $n_{atRA}$ = 33 cells in six

*Figure 1 continued on next page*

*Figure 1 continued*

samples each; RM two-way ANOVA followed by Sidak's multiple comparisons). (**D, E**) Passive membrane properties, that is, resting membrane potential (**D**) and input resistance (**E**) from atRA- and vehicle-only-treated neurons ($n_{control}$ = 43 cells, $n_{atRA}$ = 33 cells in six samples each; Mann–Whitney test). (**F, G**) Sample traces and group data of AMPA receptor-mediated spontaneous excitatory postsynaptic currents (sEPSCs; $n_{control}$ = 44 cells, $n_{atRA}$ = 33 cells in six samples each; Mann–Whitney test, U = 454 for sEPSC amplitude analysis, p=0.12 for sEPSC frequency). Individual data points are indicated by gray dots. Values represent mean ± s.e.m. (ns, non-significant difference, **p<0.01).

The online version of this article includes the following figure supplement(s) for figure 1:

**Figure supplement 1.** All-trans retinoic acid (atRA) induces excitatory synaptic strengthening in human superficial (layer 2/3) pyramidal neurons – an in-sample control analysis.

of excitatory neurotransmission onto human cortical pyramidal neurons. Since no major changes in active or passive membrane properties were detected in these initial experiments, we focused on the effects of atRA on excitatory synapses and dendritic spines.

## All-trans retinoic acid and dendritic spine morphology

A positive correlation between excitatory synaptic strength, that is, sEPSC amplitude, and dendritic spine size has been demonstrated in various animal models (*Bosch and Hayashi, 2012*; *Matsuzaki et al., 2004*). Hence, we wondered whether atRA also induces structural changes in dendritic spines from adult human cortical brain slices. To address this question, a set of recorded neurons was filled with biocytin and stained with Alexa Fluor-labeled streptavidin to visualize dendritic spine morphologies (*Figure 2A*). No significant differences in spine density were observed between the two groups (*Figure 2B*). However, marked increases in spine head sizes were evident in the atRA-treated group (*Figure 2C*). These findings identify that excitatory synaptic strength, as indicated by sEPSC amplitudes, is positively correlated with dendritic spine size in human cortical pyramidal cells, thus revealing coordinated structural and functional changes of excitatory postsynaptic membranes in atRA-treated human cortical slices.

## All-trans retinoic acid and synaptopodin

Previous work revealed that the actin-modulating molecule synaptopodin (*Mundel et al., 1997*) is an essential component of the spine apparatus organelle (*Deller et al., 2003*). This enigmatic cellular organelle is composed of stacked smooth endoplasmic reticulum and is found in subsets of dendritic spines (*Kulik et al., 2019*; *Spacek, 1985*). Synaptopodin-deficient mice lack spine apparatus organelles and exhibit defects in synaptic plasticity and behavioral learning (*Deller et al., 2003*; *Jedlicka et al., 2009*; *Vlachos et al., 2013*). In this context, we previously showed that synaptopodin promotes the accumulation of AMPA receptors at synaptic sites (*Vlachos et al., 2009*). Given that atRA and synaptopodin have been linked to AMPA receptor-mediated synaptic plasticity (*Aoto et al., 2008*; *Arendt et al., 2015a*; *Maggio and Vlachos, 2018*; *Poon and Chen, 2008*; *Vlachos et al., 2013*; *Vlachos et al., 2009*), we asked whether atRA mediates its effects via synaptopodin and the spine apparatus organelle (*Figure 2D–J*).

Synaptopodin clusters were detected in a considerable number of dendritic spines of human superficial (layer 2/3) pyramidal neurons: 74 ± 2% of all dendritic spines contained a synaptopodin cluster in a characteristic position, that is, in the base, neck, or head of the spine (*Figure 2D*). Similar to previous reports (*Holbro et al., 2009*; *Vlachos et al., 2009*; *Yap et al., 2020*), we also observed a positive correlation between synaptopodin cluster size and spine head size in the human cortex (*Figure 2E*). Finally, synaptopodin-positive spines were significantly larger than their synaptopodin-negative neighbors (*Figure 2F*). Hence, this study reveals that synaptopodin clusters are found in strategic positions in a subset of large dendritic spines of the adult human cortex.

Systematic assessment of synaptopodin-positive and synaptopodin-negative spines revealed that atRA does not act specifically on synaptopodin-containing spines (*Figure 2F*); in fact, significant atRA-mediated increases in spine head size were observed in both synaptopodin-positive and -negative dendritic spines. Although we did not observe a significant difference in the number of synaptopodin-positive spines between the two groups (control: 71 ± 4% in 14 dendritic segments; atRA: 77 ± 2% in 21 dendritic segments; p=0.25, Mann–Whitney test), atRA caused a significant increase in the sizes of synaptopodin clusters (*Figure 2G*; c.f., *Figure 2E*). We conclude that remodeling of

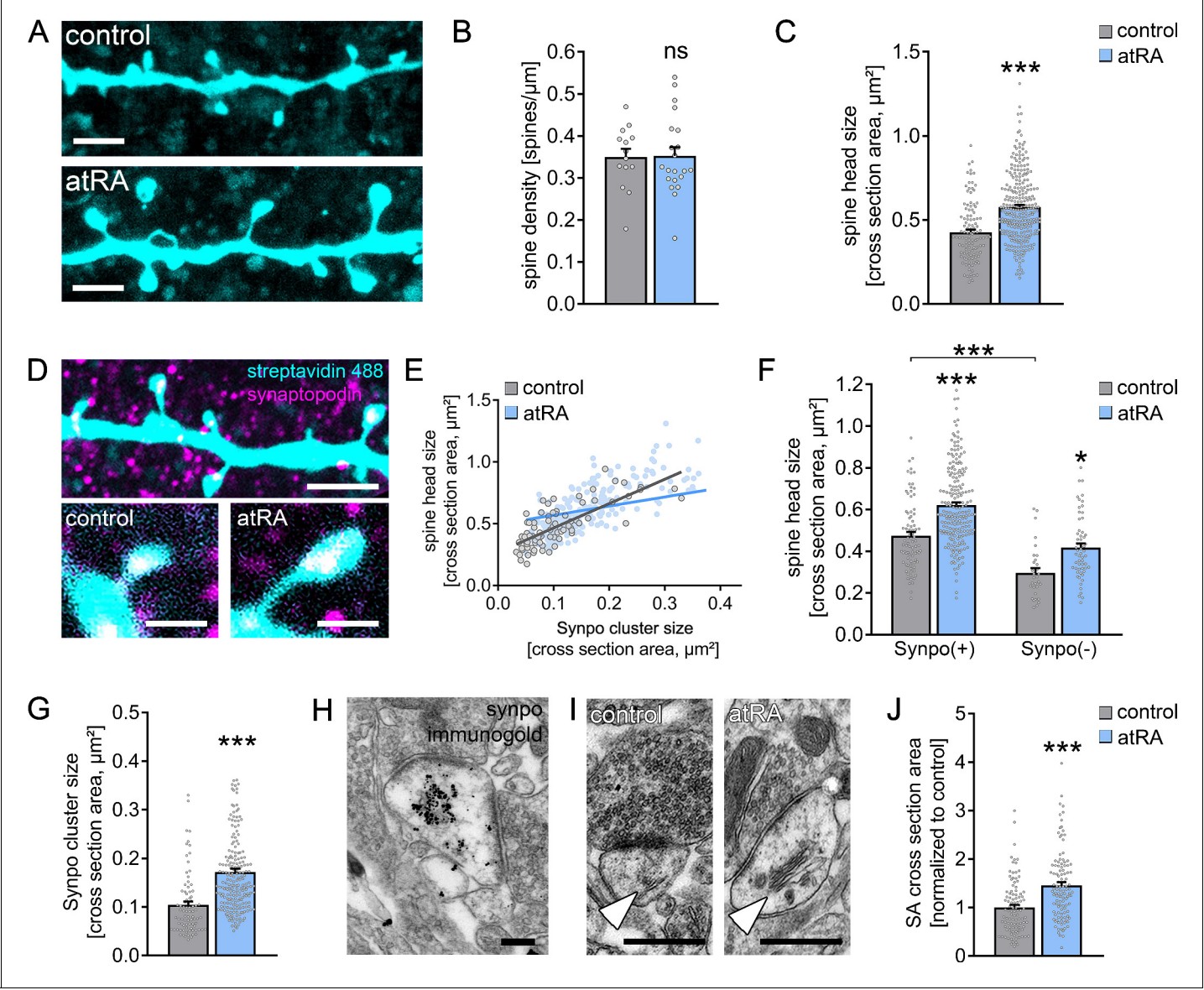

**Figure 2.** All-trans retinoic acid (atRA) induces dendritic spine plasticity in human cortical slices. (A) Example of dendritic segments of post hoc-labeled superficial (layer 2/3) pyramidal neurons in atRA- (1 µM, 6–10 hr) and vehicle-only-treated slices. Scale bars = 3 µm. (B, C) Group data for spine densities ($n_{control}$ = 14 dendritic segments, $n_{atRA}$ = 21 dendritic segments from three samples each; Mann–Whitney test) and spine head sizes ($n_{control}$ = 115 dendritic spines from 14 dendritic segments, $n_{atRA}$ = 267 from 21 dendritic segments, three samples each; Mann–Whitney test, U = 8653). (D) Representative images of synaptopodin (Synpo) stained dendritic segments. Scale bar (upper panel) = 5 µm, scale bars (lower panels) = 1 µm. (E–G) Correlation of synaptopodin cluster size and spine head size (E, $n_{control}$ = 84 dendritic spines, $n_{atRA}$ = 208 dendritic spines; Spearman r control = 0.73*** and atRA = 0.69***; three data points outside the axis limits in the atRA-treated group), group data of spine head sizes in synaptopodin-positive and -negative spines (F, synaptopodin-positive spines: $n_{control}$ = 84, $n_{atRA}$ = 208; synaptopodin-negative spines: $n_{control}$ = 31, $n_{atRA}$ = 59; Kruskal–Wallis test followed by Dunn's multiple comparisons; one data point outside the axis limits in the atRA-treated group of synaptopodin-positive spines), and synaptopodin cluster sizes in atRA- and vehicle-only-treated slices (G, $n_{control}$ = 84, $n_{atRA}$ = 208; Mann–Whitney test, U = 3992; three data points outside the axis limits in the atRA-treated group). (H) Electron micrograph of synaptopodin immunogold-labeled spine apparatus (SA). Scale bar = 250 nm. (I, J) Examples and group data of SA (white arrowheads) cross-sectional areas in atRA- and vehicle-only-treated slices ($n_{control}$ = 103, $n_{atRA}$ = 114 from three samples each; values were normalized to the mean cross-section area in the vehicle-only-treated group; Mann–Whitney test, U = 3489). Scale bar = 500 nm. Individual data points are indicated by gray dots. Values represent mean ± s.e.m. (ns, non-significant difference, *p<0.05, ***p<0.001).
The online version of this article includes the following figure supplement(s) for figure 2:

**Figure supplement 1.** Ultrastructural analysis of excitatory synapses reveals a positive correlation between sizes of the spine head and the spine apparatus organelle.

synaptopodin clusters accompanies atRA-mediated coordinated structural and functional changes in human dendritic spines.

## atRA and the spine apparatus organelle

In an earlier study, we showed that remodeling of synaptopodin clusters reflects plasticity-related ultrastructural changes in spine apparatus organelles (*Vlachos et al., 2013*). We therefore wondered whether atRA changes the ultrastructural properties of the spine apparatus organelle in human cortical neurons (*Figure 2H–J*). After confirming that synaptopodin is a marker of the human spine apparatus organelle using pre-embedding immunogold labeling techniques (*Figure 2H*), the ultrastructural properties of this organelle were assessed in cross-sectional transmission electron micrographs of 103 control and 114 atRA-treated asymmetric spine synapses from three independent samples (*Figure 2I,J*). A marked increase in the cross-sectional area of the spine apparatus organelle was observed in the atRA-treated group, which is consistent with the atRA-induced increase in synaptopodin cluster size (*Figure 2J*; positive correlations between spine apparatus cross-sections and spine cross-sections are shown in *Figure 2—figure supplement 1*).

Taken together, these results reveal a link between synaptopodin and the spine apparatus organelle in the human cortex. Specifically, these findings demonstrate that changes in the structural and functional properties of dendritic spines are accompanied by ultrastructural changes in spine apparatus organelles. We conclude that atRA is a potent mediator of coordinated (ultra)structural and functional synaptic changes in the adult human cortex.

## Effects of atRA in the neocortex of wild-type and synaptopodin-deficient mice

To further understand the relevance of synaptopodin and the spine apparatus organelle in atRA-mediated synaptic plasticity, we prepared acute medial prefrontal cortex (mPFC) slices from synaptopodin-deficient ($Synpo^{-/-}$; *Deller et al., 2003*) and age-matched wild-type mice ($Synpo^{+/+}$). Single-cell recordings of superficial (layer 2/3) pyramidal neurons from $Synpo^{+/+}$ slices showed that changes in excitatory neurotransmission are similar to what we observed in the human cortical slices: sEPSC amplitudes (but not frequencies) were significantly increased following atRA treatment (*Figure 3A,B*). Notably, atRA also reduced the input resistance of these pyramidal cells (*Figure 3—figure supplement 1*). The resting membrane potentials and action potential (AP) frequencies were not affected by atRA, similar to what we observed in the human cortical slices (*Figure 3—figure supplement 1*; c.f., *Figure 1*). We conclude that atRA exerts comparable effects on excitatory neurotransmission in both the adult mouse and human neocortex.

In $Synpo^{-/-}$ preparations, no significant changes in sEPSC properties were observed following atRA treatment (*Figure 3C,D*), thus demonstrating the relevance of synaptopodin in atRA-mediated synaptic plasticity. Additionally, a reduction in the input resistance was not observed in atRA-treated $Synpo^{-/-}$ preparations (*Figure 3—figure supplement 1*). Furthermore, the active and passive membrane properties of $Synpo^{+/+}$ and $Synpo^{-/-}$ neurons were indistinguishable (*Figure 3—figure supplement 1*), despite significant increases in the baseline sEPSC properties of $Synpo^{-/-}$ preparations (*Figure 3—figure supplement 2*).

To confirm and extend these findings, sEPSC recordings were carried out in acute slices prepared from Thy1-GFP/$Synpo^{+/-} \times Synpo^{-/-}$ mice. In this mouse model, the synaptopodin coding sequence is tagged with green fluorescent protein (GFP) and expressed under the control of the Thy1.2 promotor in the absence of endogenous synaptopodin ($Synpo^{-/-}$ genetic background; *Vlachos et al., 2013*). In a previous study, we demonstrated that the transgenic expression of GFP/Synpo rescues the ability of $Synpo^{-/-}$ neurons to form spine apparatus organelles and to express synaptic plasticity (*Vlachos et al., 2013*). Indeed, a significant increase in sEPSC amplitudes was observed in GFP/Synpo expressing pyramidal neurons following atRA treatment (*Figure 3E,F*). Taken together, we conclude that the presence of synaptopodin is required for atRA-mediated synaptic strengthening.

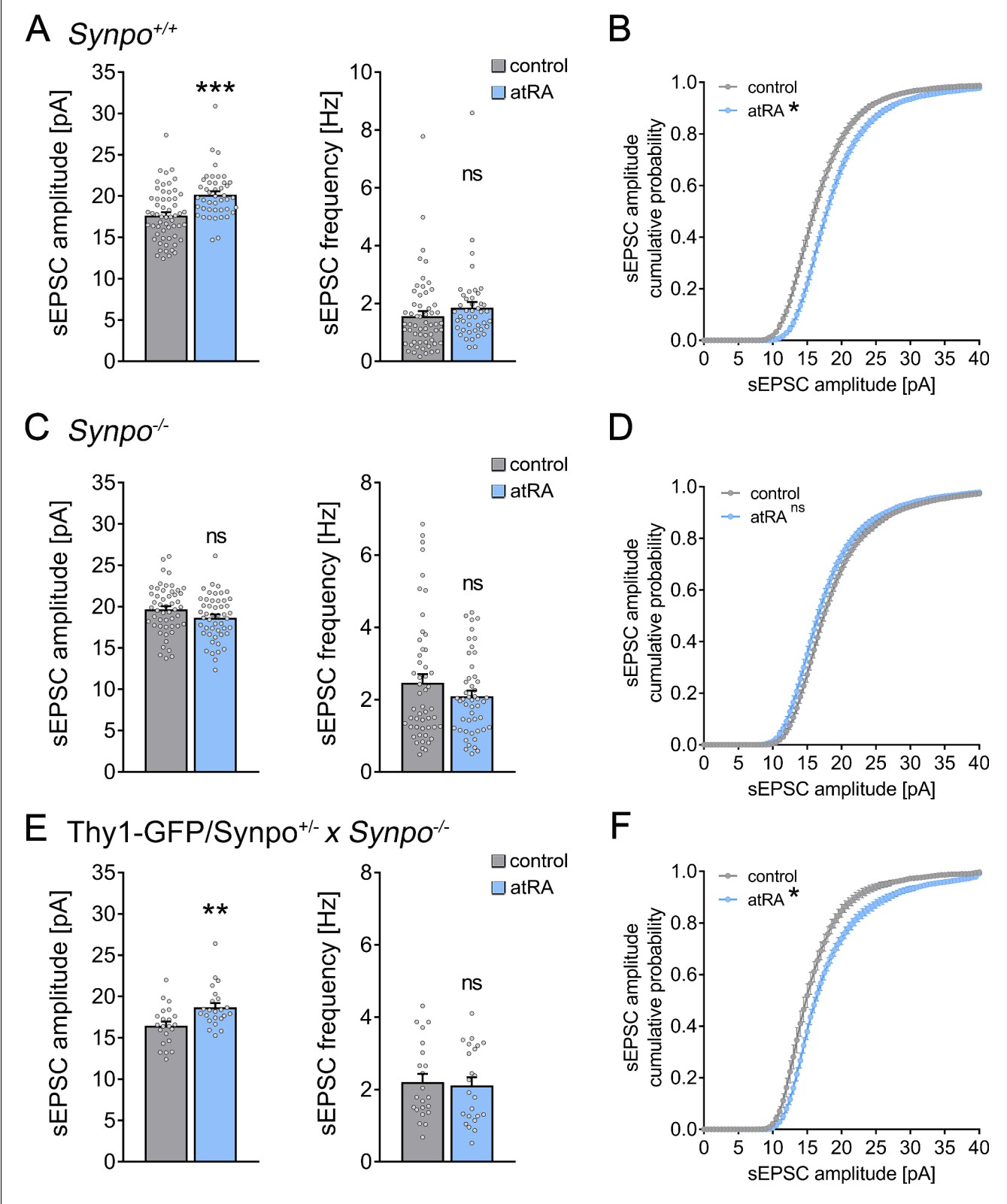

**Figure 3.** Effects of all-trans retinoic acid (atRA) in cortical slices prepared from synaptopodin-deficient mice. (**A, B**) Group data (**A**) of AMPA receptor-mediated spontaneous excitatory postsynaptic currents (sEPSCs) recorded from superficial (layer 2/3) pyramidal neurons of the dorsomedial prefrontal cortex in slices prepared from wild-type animals ($Synpo^{+/+}$) and cumulative distribution (**B**) of sEPSC amplitudes ($n_{control}$ = 58 cells, $n_{atRA}$ = 44 cells in seven independent experiments; Mann–Whitney test for column statistics, $U_{sEPSC\ amplitude}$ = 684; RM two-way ANOVA followed by Sidak's multiple

*Figure 3 continued on next page*

Figure 3 continued

comparisons for statistical evaluation of cumulative sEPSC amplitude distributions). (C, D) Group data (C) of AMPA receptor-mediated sEPSCs recorded from superficial (layer 2/3) pyramidal neurons of the dorsomedial prefrontal cortex in slices prepared from synaptopodin-deficient mice (Synpo$^{-/-}$) and cumulative distribution (D) of sEPSC amplitudes ($n_{control}$ = 51 cells, $n_{atRA}$ = 49 cells in seven independent experiments; Mann–Whitney test for column statistics and RM two-way ANOVA followed by Sidak's multiple comparisons for statistical evaluation of cumulative sEPSC amplitude distributions). (E, F) Group data (E) of sEPSC recordings and cumulative distribution (F) of sEPSC amplitudes in cortical slices prepared from transgenic mice expressing GFP-tagged synaptopodin under the control of the Thy1.2 promotor on synaptopodin-deficient genetic background (Thy1-GFP/Synpo$^{+/-}$ x Synpo$^{-/-}$; $n_{control}$ = 22 cells, $n_{atRA}$ = 23 cells in three independent experiments; Mann–Whitney test for column statistics, $U_{sEPSC\ amplitude}$ = 125; RM two-way ANOVA followed by Sidak's multiple comparisons for statistical evaluation of cumulative sEPSC amplitude distributions). Individual data points are indicated by gray dots. Values represent mean ± s.e.m. (ns, non-significant difference, ***p<0.001, **p<0.01).

The online version of this article includes the following figure supplement(s) for figure 3:

**Figure supplement 1.** Analysis of intrinsic cellular properties from superficial pyramidal neurons in wild-type and synaptopodin-deficient slices upon atRA treatment.

**Figure supplement 2.** Comparison of baseline spontaneous excitatory synaptic transmission in wild-type, synaptopodin-deficient, and transgenic GFP/Synpo superficial pyramidal neurons of the medial prefrontal cortex.

## Pharmacologic inhibition of gene transcription and mRNA translation in murine cortical slices

The biological effects of atRA occur at the levels of gene transcription and mRNA translation (i.e., protein synthesis; *Drager, 2006*; *Poon and Chen, 2008*). Therefore, in a different set of acute cortical slices prepared from wild-type mice, actinomycin D (5 µg/ml) was used to block gene transcription in the presence of atRA (1 µM; 6–10 hr; *Figure 4A,B*). To control for possible toxic effects, the basic active and passive membrane properties of cortical slices treated with actinomycin D or vehicle-only were recorded (*Figure 4—figure supplement 1*). Other than a reduction in AP frequencies at high-current injections, we did not observe any significant differences between the two groups (*Figure 4—figure supplement 1*). Moreover, baseline sEPSC properties were indistinguishable between actinomycin D- and vehicle-only-treated slices (*Figure 4A*). However, we observed a significant increase in the amplitudes of AMPA receptor-mediated sEPSCs in the atRA-treated group in both vehicle-only and actinomycin D co-incubated slices (*Figure 4A,B*), thus further confirming that atRA mediates synaptic strengthening. Notably, the atRA-mediated reduction in input resistance was not observed in these experiments (*Figure 4—figure supplement 1*).

We next used anisomycin (10 µM) to block mRNA translation during atRA treatment (*Figure 4C, D*). Anisomycin incubation had no major effects on passive membrane properties, while reductions in AP frequencies were observed at high-current injections (*Figure 4—figure supplement 1*). In contrast to our experiments with actinomycin D (*Figure 4A*), however, no significant changes in sEPSC amplitudes were detected in the presence of both atRA and anisomycin (*Figure 4C,D*). Together, these results indicate that the effects of atRA on excitatory neurotransmission do not require major transcriptional changes, but depend on mRNA translation and protein synthesis.

## Pharmacologic inhibition of mRNA translation in human cortical slices

Based on the results we obtained in cortical slices from the murine brain, we returned to our human brain slice model to evaluate whether mRNA translation is required for atRA-mediated structural and functional synaptic plasticity in the human cortex (*Figure 5*). We recorded AMPA receptor-mediated sEPSCs once again from superficial (layer 2/3) pyramidal neurons of atRA- and vehicle-treated human cortical slices in the presence of the pharmacologic mRNA translation inhibitor anisomycin (10 µM; 6–10 hr). No significant differences in sEPSC amplitudes were observed between the two groups (*Figure 5A,B*). Consistent with these findings, anisomycin also blocked the observed atRA-mediated increases in dendritic spine head and synaptopodin cluster sizes, whereas the sizes of synaptopodin-positive and synaptopodin-negative spine heads remain significantly different (*Figure 5C, D*). These results demonstrate that atRA-mediated plasticity requires mRNA translation to trigger coordinated changes in synaptic strength, spine head size, and synaptopodin cluster properties in cortical slices prepared from the adult human brain.

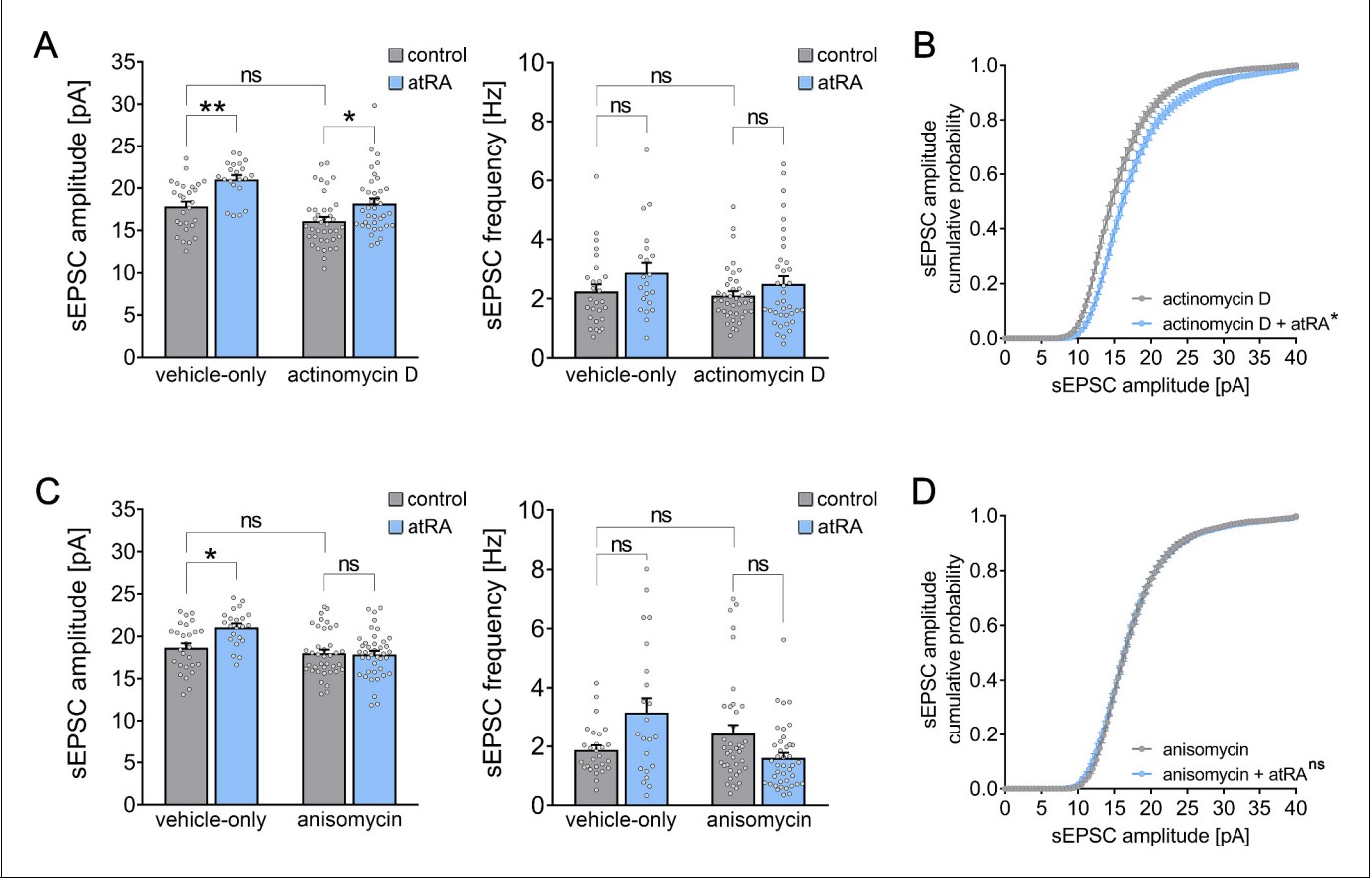

**Figure 4.** All-trans retinoic acid (atRA)-induced strengthening of excitatory synapses depends on mRNA translation, but not gene transcription. (A, B) Group data (A) of AMPA receptor-mediated spontaneous excitatory postsynaptic currents (sEPSCs) recorded from superficial (layer 2/3) pyramidal neurons of the dorsomedial prefrontal cortex in slices prepared from wild-type mice treated with atRA in the presence of actinomycin D (5 µg/ml) or vehicle-only (vehicle-only: $n_{control}$ = 27 cells, $n_{atRA}$ = 21 cells in four independent experiments; actinomycin D: $n_{control}$ = 39 cells, $n_{atRA}$ = 37 cells in six independent experiments; Kruskal–Wallis test followed by Dunn's multiple comparisons). Cumulative distribution (B) of sEPSC amplitudes in actinomycin D co-incubated slices confirms the atRA-induced strengthening of spontaneous excitatory neurotransmission (B; RM two-way ANOVA followed by Sidak's multiple comparisons). (C, D) Group data (C) of AMPA receptor-mediated sEPSCs recorded from superficial (layer 2/3) pyramidal neurons of the dorsomedial prefrontal cortex in slices prepared from wild-type mice treated with atRA in the presence of anisomycin (10 µM) or vehicle-only (vehicle-only: $n_{control}$ = 27 cells, $n_{atRA}$ = 22 cells in four independent experiments; anisomycin: $n_{control}$ = 38 cells, $n_{atRA}$ = 40 cells in six independent experiments; Kruskal–Wallis test followed by Dunn's multiple comparisons). Cumulative distribution (D) of sEPSC amplitudes in anisomycin co-incubated slices confirms that anisomycin blocks atRA-mediated plasticity at excitatory synapses (RM two-way ANOVA followed by Sidak's multiple comparisons). Individual data points are indicated by gray dots. Values represent mean ± s.e.m. (ns, non-significant difference, *p<0.05, **p<0.01). The online version of this article includes the following figure supplement(s) for figure 4:

**Figure supplement 1.** Intrinsic cellular properties of superficial pyramidal neurons upon atRA treatment and simultaneous pharmacological inhibition of either gene transcription or mRNA translation.

## Discussion

Vitamin A and its metabolites bind to nuclear and cytoplasmic receptors and regulate important biological processes, such as cell growth, cell survival, and differentiation (*Al Tanoury et al., 2013*). The pleiotropic effects of these molecules account for both the therapeutic (e.g., for treating promyelocytic leukemia) and teratogenic effects of atRA (*Hu et al., 2009*). Studies of the role of atRA in the central nervous system have primarily focused on embryonic and early postnatal development (*Luo et al., 2004*; *Rataj-Baniowska et al., 2015*). However, a growing body of literature indicates that physiological retinoid metabolism and signaling also occur in the adult murine brain (*Shearer et al., 2012*). Several animal studies have revealed that retinoid receptors bound to specific cytoplasmic mRNAs control synaptic plasticity by regulating the synthesis, trafficking, and

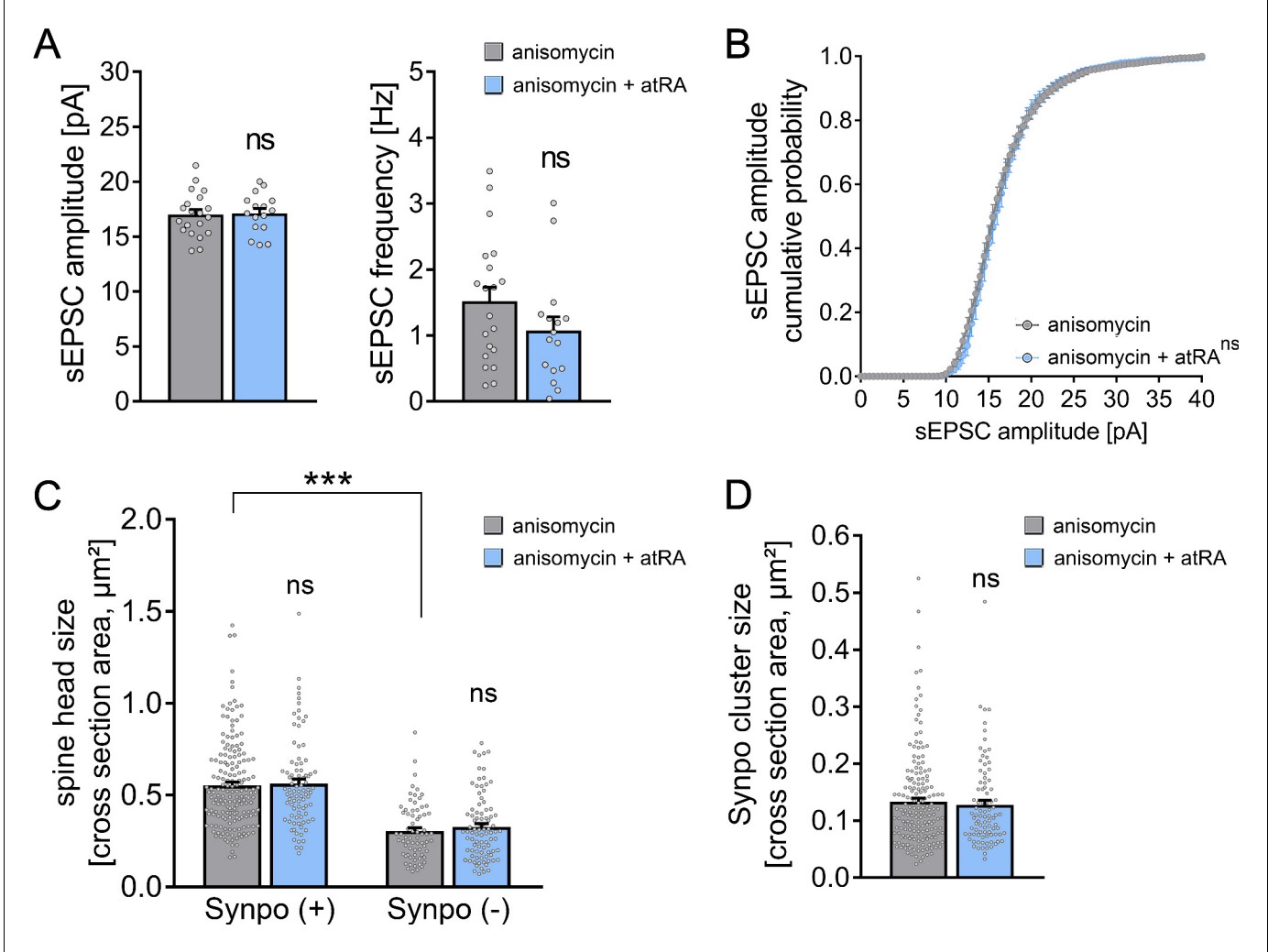

**Figure 5.** Pharmacologic inhibition of mRNA translation prevents all-trans retinoic acid (atRA)-induced synaptic plasticity in human cortical slices. (A, B) Group data (A) of AMPA receptor-mediated spontaneous excitatory postsynaptic currents (sEPSCs) and cumulative distribution (B) of sEPSC amplitudes recorded from superficial (layer 2/3) pyramidal neurons in adult human cortical slices treated with atRA (1 µM, 6–10 hr) or vehicle-only in the presence of anisomycin (10 µM; $n_{control}$ = 20 cells, $n_{atRA}$ = 16 cells in three independent experiments; Mann–Whitney test for column statistics; RM two-way ANOVA followed by Sidak's multiple comparisons for statistical evaluation of cumulative sEPSC amplitude distribution). (C, D) Spine head sizes (C) and synaptopodin cluster sizes (D) in human cortical slices treated with atRA or vehicle-only in the presence of anisomycin (c.f., **Figure 2F,G**; synaptopodin-positive spines: $n_{control}$ = 175, $n_{atRA}$ = 88; synaptopodin-negative spines: $n_{control}$ = 69, $n_{atRA}$ = 88, 1–11 segments per sample in three independent experiments; Kruskal–Wallis test followed by Dunn's multiple comparisons for spine head size and Mann–Whitney test for synaptopodin cluster size). Individual data points are indicated by gray dots. Values represent mean ± s.e.m. (ns, non-significant difference, ***p<0.001).

accumulation of synaptic proteins (*Aoto et al., 2008*; *Groth and Tsien, 2008*; *Maghsoodi et al., 2008*; *Poon and Chen, 2008*). Some of these findings have recently been replicated in neurons derived from human iPSCs (*Zhang et al., 2018*). The results of the present study demonstrate the ability of atRA to induce synaptic plasticity in the adult human cortex. Consistent with recent reports on the importance of protein synthesis in synaptic plasticity (*Biever et al., 2020*; *Hafner et al., 2019*; *Sutton and Schuman, 2006*), our experiments revealed that atRA-mediated structural and functional synaptic plasticity in adult human cortical slices requires mRNA translation.

At the mechanistic level, we identified synaptopodin as a target and mediator of atRA-induced synaptic plasticity. Synaptopodin is an actin-modulating protein (*Mundel et al., 1997*) that has been linked to the spine apparatus organelle (*Deller et al., 2003*), a membranous extension of the smooth endoplasmic reticulum found in a subpopulation of telencephalic dendritic spines (*Gray, 1959*; *Spacek, 1985*; *Spacek and Harris, 1997*). A critical role of synaptopodin in the formation of the

spine apparatus was reported in $Synpo^{-/-}$ mice (*Deller et al., 2003*); the neurons of these mice do not form spine apparatus organelles, and $Synpo^{-/-}$ animals exhibit defects in synaptic plasticity and memory formation (*Deller et al., 2003*; *Jedlicka et al., 2009*; *Korkotian et al., 2014*; *Maggio and Vlachos, 2018*; *Vlachos et al., 2013*; *Vlachos et al., 2009*). The results of the present study demonstrate that (1) approximately 70% of dendritic spines in the superficial layers of the human cortex contain synaptopodin clusters; (2) synaptopodin is a marker of the human spine apparatus organelle; (3) synaptopodin clusters and spine apparatus organelles are found in large dendritic spines; and (4) a plasticity-inducing stimulus, that is, application of atRA, promotes remodeling of synaptopodin clusters, spine apparatus organelles, and dendritic spines in cortical slices prepared from the adult human brain (c.f., *Figure 2*).

The involvement of synaptopodin and the spine apparatus organelle in synaptic plasticity is still enigmatic; however, a role in local protein synthesis and the regulation of intracellular calcium dynamics have been suggested (*Fifková et al., 1983*; *Pierce et al., 2000*). Synaptopodin binds to actin and α-actinin and has been suggested to orchestrate spine dynamics via Rho-A signaling and long-term spine stability (*Asanuma et al., 2006*; *Yap et al., 2020*). More recently, synaptopodin and myosin V were reported to be associated (*Konietzny et al., 2019*). Thus, synaptopodin may be relevant for the parallel functional and structural changes observed at excitatory synapses undergoing glutamate receptor-mediated synaptic plasticity (*Jedlicka and Deller, 2017*; *Vlachos et al., 2009*). However, the strategic positioning of synaptopodin clusters in the head, neck, and base of murine and human dendritic spines indicates that this molecule may act primarily within or in association with the spine compartment. Here, synaptopodin may link spine actin and myosin to spine apparatus organelles, which have a more restricted localization (*Konietzny et al., 2019*). In support of this hypothesis, atRA application in our study triggered changes in spine head size in both synaptopodin-positive and synaptopodin-negative spines from human cortical slices, suggesting that the rapid/initial atRA-mediated spine enlargement does not require the actual presence of synaptopodin clusters, that is, spine apparatus organelles (*Okubo-Suzuki et al., 2008*). Considering that synaptopodin clusters are highly dynamic structures that can change their position within spines, and may appear in existing spines (*Vlachos et al., 2009*; *Yap et al., 2020*), it is interesting to speculate that atRA may trigger a molecular cascade that promotes rapid spine growth, even in spines that do not contain a spine apparatus organelle, eventually leading to the formation and enlargement of spine apparatus organelles. However, more work is required to identify the precise downstream signaling pathways through which atRA mediates synaptopodin- and protein synthesis-dependent synaptic plasticity in the adult human cortex. Similarly, the role of synaptopodin in atRA-mediated changes in intrinsic cellular properties, for example, reductions in the input resistance of principal neurons in the mouse cortex, warrants further investigation, particularly in the context of recent work on the distinct electrical properties of human and murine cortical neurons (*Gidon et al., 2020*).

Both atRA and synaptopodin have been linked to the expression of Hebbian and homeostatic synaptic plasticity in rodent brain tissue and are associated with $Ca^{2+}$-dependent AMPA receptor synthesis and trafficking (*Arendt et al., 2015b*; *Vlachos et al., 2013*; *Vlachos et al., 2009*). In turn, alterations in retinoid signaling and synaptopodin expression have been associated with synaptic plasticity defects in pathological brain states (*Bremner et al., 2012*; *Maggio and Vlachos, 2014*). Considering that increased sEPSC amplitudes and frequencies were observed in the cortical neurons of $Synpo^{-/-}$ mice, it would be interesting to evaluate whether alterations in retinoid metabolism are present in this mouse model. Accordingly, alterations in synaptopodin expression and retinoid signaling have been observed in the brain tissue of patients with Alzheimer's disease and cognitive decline (*Goodman, 2006*; *Goodman and Pardee, 2003*; *Mingaud et al., 2008*; *Misner et al., 2001*; *Reddy et al., 2005*). Given that retinoids have been proposed as a potential therapeutic avenue for Alzheimer's disease-associated cognitive decline (*Ding et al., 2008*; *Endres et al., 2014*), it is conceivable that atRA may act – at least in part – by modulating synaptopodin expression, thereby improving the ability of adult human cortical neurons to express synaptic plasticity. We are confident that the translational approach used in this study investigating both murine and human cortical slices will facilitate the identification of key mechanisms of synaptic plasticity in the adult human brain.

# Materials and methods

## Key resources table

| Reagent type (species) or resource | Designation | Source or reference | Identifiers | Additional information |
|---|---|---|---|---|
| Antibody | Anti-Synaptopodin (Rabbit polyclonal) | Synaptic Systems | Cat#: 163002 RRID:AB_887825 | IF ('1:1000') EM ('1:100') |
| Antibody | Anti-NeuN (Rabbit polyclonal) | Abcam | Cat#: ab104225 RRID:AB_10711153 | IF ('1:500') |
| Antibody | Anti-Rabbit IgG (H+L) Highly Cross-Adsorbed Secondary Antibody, Alexa Fluor 488 (Goat polyclonal) | Invitrogen | Cat#: A-11034, RRID:AB_2576217 | IF ('1:1000') |
| Antibody | Anti-Rabbit IgG (H+L) Highly Cross-Adsorbed Secondary Antibody, Alexa Fluor Plus 555 (Goat polyclonal) | Invitrogen | Cat#: A-32732, RRID:AB_2633281 | IF ('1:1000') |
| Antibody | Anti-Rabbit IgG Nanogold-Fab' (goat polyclonal) | Nanoprobes | Cat#: 2004 RRID:AB_2631182 | EM ('1:100') |
| Biological sample (*Homo sapiens*), male and female | Sample | Biobank of the Department for Neurosurgery at the Faculty of Medicine, University of Freiburg, AZ 472/15_160880 | | Approval of the Local Ethics Committee, University of Freiburg, AZ 593/19 |
| Chemical compound, drug | DAPI (1 mg/ml in water) | Thermo Scientific | Cat#: 62248 | IF and post hoc labeling ('1:5000') |
| Chemical compound, drug | Pierce16% Formaldehyde (w/v), methanol-free | Thermo Scientific | Cat#: 28906 | Final concentration: (4% in PBS) |
| Chemical compound, drug | Glutardialdehyd | Carl Roth | Cat#: 4157.2 | Final concentration: 2.5% (TEM) and 0.1% (Immunogold) |
| Chemical compound, drug | All-trans retinoic acid | Sigma–Aldrich | Cat#: R2625 | Final concentration: 1 µM |
| Chemical compound, drug | Anisomycin | Abcam | Cat#: ab120495 | Final concentration: 10 µM |
| Chemical compound, drug | Actinomycin D | Sigma–Aldrich | Cat#: A9415 | Final concentration: 5 µg/ml |
| Commercial assay, kit | HQ Silver Enhancement Kit | Nanoprobes | Cat#: 2012 | |
| Genetic reagent *Mus musculus*, male | B6.129-Synpo^tm1Mndl/Dllr; *Synpo*$^{-/-}$ | *Vlachos et al., 2013* PMID:23630268 | MGI: 6423115 | Obtained from Deller Lab (Frankfurt) |
| Genetic reagent *Mus musculus*, male | B6.Cg-Synpo^tm1Mndl Tg(Thy1-Synpo/GFP)1Dllr/Dllr; Thy1-GFP/Synpo$^{+/-}$ x Synpo$^{-/-}$ | *Vlachos et al., 2013* PMID:23630268 | MGI: 6423116 | Obtained from Deller Lab (Frankfurt) |
| Peptide, recombinant protein | Streptavidin, Alexa Fluor 488-Conjugate | Invitrogen | Cat#: S32354 RRID:AB_2315383 | Post hoc labeling ('1:1000') |
| Software, algorithm | Prism | GraphPad | RRID:SCR_002798 | |
| Software, algorithm | Clampfit (pClamp software package) | Molecular Devices | RRID:SCR_011323 | |
| Software, algorithm | ImageJ | | RRID:SCR_003070 | |

*Continued on next page*

*Continued*

| Reagent type (species) or resource | Designation | Source or reference | Identifiers | Additional information |
|---|---|---|---|---|
| Software, algorithm | Photoshop | Adobe | RRID:SCR_014199 | |
| Strain, strain background *Mus musculus* | C57BL/6J; *Synpo*$^{+/+}$ | Jackson Laboratory | RRID: IMSR_JAX:000664 | |

## Preparation of acute human cortical slices

After resection, cortical access tissue was immediately transferred to an oxygenated extracellular solution containing (in mM) 92 NMDG, 2.5 KCl, 1.25 $NaH_2PO_4$, 30 $NaHCO_3$, 20 HEPES, 25 glucose, 2 thiourea, 5 Na-ascorbate, 3 Na-pyruvate, 0.5 $CaCl_2$, and 10 $MgSO_4$, (pH = 7.3–7.4) at approximately 10°C (NMDG-aCSF; *Gidon et al., 2020*; *Ting et al., 2018*). Prior to slicing, cortical tissue was embedded in low-melting-point agarose (Sigma–Aldrich, #A9517; 1.8% [w/v] in phosphate-buffered saline [PBS]). Tissue sections (400 µm) were cut with a Leica VT1200S vibratome perpendicular to the pial surface in the same solution at 10°C under continuous oxygenation (5% $CO_2$/95% $O_2$). Slices were transferred to cell strainers with 40 µm pores and placed in NMDG-aCSF at 34°C. Subsequently, sodium levels were gradually increased as previously described (*Ting et al., 2018*). After recovery, slices were maintained for further experimental assessment at room temperature in an extracellular solution containing (in mM) 92 NaCl, 2.5 KCl, 1.25 $NaH_2PO_4$, 30 $NaHCO_3$, 20 HEPES, 25 glucose, 2 thiourea, 5 Na-ascorbate, 3 Na-pyruvate, 2 $CaCl_2$, and 2 $MgSO_4$. Cortical slices from all human samples were macroscopically normal and showed no overt pathology.

## Preparation of acute mouse cortical slices

Adult mice (C57BL/6J, B6.129-Synpo$^{tm1Mndl}$/Dllr [referred to as *Synpo*$^{-/-}$] and B6.Cg-Synpo$^{tm1Mndl}$Tg (Thy1-Synpo/GFP)1Dllr/Dllr [referred to as Thy1-GFP/Synpo$^{+/-}$× *Synpo*$^{-/-}$]; 6–11 weeks old) were used in this study. In *Synpo*$^{-/-}$ animals, the synaptopodin coding sequence is replaced by the lacZ sequence (encoding β-galactosidase), thereby achieving a null synaptopodin allele (*Deller et al., 2003*). This mouse strain was backcrossed onto the C57BL/6 genetic background for at least 10 generations. In experiments involving synaptopodin-deficient preparations, age-matched C57BL/6J animals served as controls, and experimental findings were confirmed using wild-type littermates from *Synpo*$^{-/-}$ mice. For preparing acute slices, animals were anesthetized with isoflurane and rapidly decapitated. Brains were rapidly removed, washed in chilled (approximately 10°C) NMDG-aCSF, and embedded in low-melting-point agarose (Sigma–Aldrich #A9517; 1.8% w/v in PBS). Coronal sections of the mPFC were prepared using a Leica VT1200S vibratome in NMDG-aCSF with the brain tilted dorsally at a 15° angle. Slice recovery and maintenance prior to experimental assessment were performed as described above for acute human cortical slices.

## Pharmacology

Acute cortical slices prepared from individual brain samples were randomly assigned to atRA or control (vehicle-only) treatment groups. Treatment with atRA was performed after slice recovery by adding atRA (1 µM, Sigma–Aldrich, #R2625) to the extracellular holding solution at a final concentration of 0.05% (v/v in DMSO). The control group from the same set of slices was handled identically but treated with vehicle-only (DMSO). Anisomycin (10 µM, Abcam, #ab120495) and actinomycin D (5 µg/ml, Sigma–Aldrich, #A9415) were added to the holding solution 10 min before the addition of atRA. Sections were treated for at least 6 hr before experimental assessment.

## Whole-cell patch-clamp recordings

Whole-cell patch-clamp recordings of superficial (layer 2/3) cortical pyramidal neurons were carried out at 35°C in a bath solution containing (in mM) 92 NaCl, 2.5 KCl, 1.25 $NaH_2PO_4$, 30 $NaHCO_3$, 20 HEPES, 25 glucose, 2 thiourea, 5 Na-ascorbate, 3 Na-pyruvate, 2 $CaCl_2$, and 2 $MgSO_4$. For experiments with acute mouse brain slices, superficial (layer 2/3) pyramidal cells in the dorsomedial prefrontal cortex were visually identified using an LN-Scope (Luigs and Neumann, Ratingen, Germany)

equipped with infrared dot-contrast and a 40× water-immersion objective (numerical aperture [NA] 0.8; Olympus). For experiments with human cortical slices, superficial (layer 2/3) pyramidal cells were visually identified on the pia-white matter axis at a distance of 500–1000 µm from the pial surface. Electrophysiological signals were amplified using a Multiclamp 700B amplifier, digitized with a Digidata 1550B digitizer, and visualized with the pClamp 11 software package. For sEPSC and intrinsic cellular property recordings, patch pipettes (tip resistance: 3–5 MΩ) contained (in mM) 126 K-gluconate, 4 KCl, 10 HEPES 4 MgATP, 0.3 $Na_2GTP$, 10 PO-creatine, and 0.3% (w/v) biocytin (pH = 7.25 with KOH; 285 mOsm/kg). For sEPSC recordings, pyramidal neurons were held at −70 mV in voltage-clamp mode. To record intrinsic cellular properties in current-clamp mode, a pipette capacitance of 2.0 pF was corrected, and series resistance was compensated using the automated bridge balance tool of the Multiclamp commander. Input–output (I-V) curves were generated by injecting 1-s square pulse currents starting at −100 pA and increasing in 10 pA increments (sweep duration: 2 s). Series resistance was monitored, and recordings were discarded if the series resistance reached >30 MΩ.

One superficial (layer 2/3) cell in human neocortical slices (*Figure 1*; atRA group) with sEPSC amplitude = 38.4 pA and sEPSC frequency = 6.5 Hz showed interneuron characteristics and was therefore excluded from the analysis. The series resistance of one cell from a human cortical slice (*Figure 1*; control group) exceeded 30 MΩ during I-V curve recording. The respective I-V-curve was therefore excluded from further analysis. In five human superficial pyramidal neurons (*Figure 1*), the number of sweeps in the I-V curve recordings was lower compared to other recordings (40 sweeps vs. 60 sweeps). Thus, cells were excluded from further analysis of action potential frequency. Furthermore, one I-V-curve recording in the actinomycin-only treated group became unstable during the last sweeps and was consecutively excluded from action potential frequency analysis (*Figure 4—figure supplement 1*). In addition, one cell from a mouse neocortical slice (*Figure 3—figure supplement 1*; *Synpo$^{+/+}$*, atRA group) was excluded from further analysis because the signal displayed a marked electrical interference that caused disturbances in the baseline of the recordings. In the same data set, I-V curve recording from one cell (*Figure 3—figure supplement 1*, *Synpo$^{+/+}$*, control group) was excluded due to I-V duplication in each sweep. Finally, one whole-cell patch-clamp recording of intrinsic cellular properties (*Figure 3—figure supplement 1*; Thy1-GFP/Synpo, control group) lost its integrity during the recording and was therefore excluded from further analysis.

## Immunostaining, post hoc labeling, and confocal microscopy

Cortical slices were fixed in 4% paraformaldehyde (PFA) (prepared from 16% PFA stocks in PBS according to the manufacturer's instructions; Thermo Scientific, #28908) at room temperature and stored at 4°C overnight in the same solution. After fixation, slices were washed in PBS and incubated for 1 hr with 10% (v/v) normal goat serum (NGS; diluted in 0.5% [v/v] Triton X-100/PBS) to reduce non-specific staining and increase antibody penetration. Subsequently, slices were incubated overnight at 4°C with rabbit anti-synaptopodin (Synaptic Systems, #163002; 1:1000) or rabbit anti-NeuN (Abcam, #ab104225, 1:500) antibodies; both antibodies were diluted in 10% (v/v) NGS in 0.1% (v/v) Triton X-100/PBS. Sections were washed with PBS and incubated with goat anti-rabbit Alexa Fluor 488 or goat anti-rabbit Alexa Fluor plus 555 labeled secondary antibodies (Invitrogen, #A-11034 and #A-32732, respectively) overnight at 4°C; both secondary antibodies were diluted 1:1000 in 10% (v/v) NGS in 0.1% (v/v) Triton X-100/PBS. For visualizing patched pyramidal cells, streptavidin-Alexa Fluor 488 (Invitrogen, #S32354; 1:1000) was added during the secondary antibody incubation. Sections were washed again and incubated for 10 min with Sudan Black B (0.1% [w/v] in 70% ethanol) to reduce autofluorescence. Sections were then incubated with DAPI for 10 min (Thermo Scientific, #62248; 1:5000 in PBS) to facilitate visualization of cytoarchitecture. After the final washing step, sections were transferred onto glass slides and mounted with a fluorescence antifade mounting medium (DAKO Fluoromount).

Confocal images were acquired using a Leica SP8 laser-scanning microscope equipped with a 20× multi-immersion (NA 0.75; Leica), a 40× oil-immersion (NA 1.30; Leica), and a 63× oil-immersion objective (NA 1.40; Leica). Image stacks for dendritic spine and synaptopodin cluster analyses were acquired with a 63× objective at 6× optical zoom (resolution: 1024 × 1024; z-step size: 0.2 µm; ideal Nyquist rate). Laser intensity and detector gain were set to achieve comparable overall fluorescence intensities throughout stacks between all groups. Confocal image stacks and single-plane pictures were stored as TIF files.

## Immunogold labeling of synaptopodin

Acute human cortical slices (400 μm) were fixed with microwave irradiation (Privileg 8020 E, 640 Watt, 2.45 GHz; *Jensen and Harris, 1989*) for 8 s on a petri dish filled with ice in 0.1% glutaraldehyde and 4% PFA (dissolved in 0.1 M phosphate buffer [PB] and 0.05 M sucrose). The slices were kept in the same solution for 1 hr at room temperature and then transferred to 0.1 M PB. After 3 hr, 50 μm sections were prepared using a Leica VT1000S vibratome, washed for 30 min in 50 mM Tris-buffered saline (TBS), and incubated for 1 hr with 20% NGS (v/v) in 50 mM TBS. Subsequently, sections were incubated with rabbit anti-synaptopodin (Synaptic Systems, #163002; 1:100 in 2% NGS/50 mM TBS [v/v]) at 4°C overnight. Sections were washed for 1 hr in 50 mM TBS and incubated with a suitable secondary goat anti-rabbit antibody (Nanoprobes, #2004; 1.4 nM gold-coupled, 1:100 in 2% NGS/50 mM TBS [v/v]) at 4°C overnight. After washing for 30 min in 50 mM TBS, sections were post-fixed with 1% glutaraldehyde/25 mM PBS (w/v) for 10 min. Sections were washed again in PBS, and silver intensification (HQ Silver Enhancement Kit, Nanoprobes, #2012) was performed according to the manufacturer's instructions. Subsequently, slices were incubated with 0.5% osmium tetroxide for 40 min, washed in graded ethanol (up to 50% [v/v]) for 10 min each, and incubated with uranyl acetate (1% [w/v] in 70% [v/v] ethanol) for 35 min. Slices were then dehydrated in graded ethanol (80%, 90%, 95%, 2× 100% for 10 min each). Two 15 min washing steps in propylene oxide were performed prior to incubation with durcupan/propylene oxide (1:3 for 45 min followed by 3:1 for 45 min) and durcupan (overnight at room temperature). After slices were embedded in durcupan, ultra-thin sectioning (55 nm) was performed using a Leica UC6 Ultracut. Sections were mounted onto copper grids (Plano), at which point an additional Pb-citrate contrasting step was performed (3 min). Electron micrographs were captured using a Philips CM100 microscope equipped with a Gatan Kamera Orius SC600 (magnification 5200x). Acquired images were stored as TIF files.

## Electron microscopy

After 6 hr of treatment with atRA or vehicle-only control, slices were fixed in 4% PFA (w/v) and 2.5% glutaraldehyde (w/v; PBS) overnight. After fixation, slices were washed for 4 hr in 0.1 M PB. Subsequently, slices were incubated with 1% osmium tetroxide for 45 min, washed in graded ethanol (up to 50% [v/v]) for 5 min each, and incubated overnight with uranyl acetate (1% [w/v] in 70% [v/v] ethanol) overnight. Slices were then dehydrated in graded ethanol (80%, 90%, 98% for 5 min each, 2× 100% for 10 min each). Subsequently, two washing steps in propylene oxide for 10 min each were performed prior to incubation with durcupan/propylene oxide (1:1 for 1 hr) and transfered to durcupan (overnight at room temperature). Slices were embedded in durcupan, and ultra-thin sectioning (55 nm) was performed using a Leica UC6 Ultracut. Sections were mounted onto copper grids (Plano), at which point an additional Pb-citrate contrasting step was performed (3 min). Electron microscopy was performed using a Philips CM100 microscope equipped with a Gatan Orius SC600 camera at 3900× magnification. Acquired images were saved as TIF files and analyzed by an investigator blinded to experimental conditions.

## Quantification and statistics

Electrophysiological data were analyzed using Clampfit 11 from the pClamp11 software package (Molecular Devices). sEPSC properties were analyzed using the automated template search tool for event detection. Specifically, AP detection was performed using the input/output curve threshold search event detection approach, whereas AP frequency was determined based upon the number of APs detected during a given injection. Artifacts in electrophysiological recordings were excluded from further analysis. Immuno-labeled synaptopodin clusters in superficial (layer 2/3) pyramidal cells of the human cortex were analyzed in image stacks of second- and third-order dendritic branches. Synaptopodin clusters that colocalized with dendritic spines (either spine neck or head; *Figure 2D*) were explored in further analyses. Single-plane images of both synaptopodin-positive and -negative clusters were extracted from image stacks at the point of maximum spine head cross-sectional area and stored as TIF files. Blinded analyses of spine head cross-sectional area and synaptopodin cluster size were performed manually using the ImageJ software package (available at http://imagej.nih.gov/ij/). Here, the outer borders of synaptopodin clusters and spine heads were marked independently of their overall fluorescence intensity. Data were transferred and stored in Excel files. Ultrastructural analysis of spine apparatus organelles was performed using single-plane images of human

cortical excitatory synapses where pre- and postsynaptic structures could be readily identified. The cross-sectional areas of spine apparatus organelles were determined manually using the ImageJ software package, independent of their shape and internal structural organization.

Statistical analyses were performed using the GraphPad Prism seven software package. Two-group comparisons were performed using a Mann–Whitney U test; U-values for statistically significant differences are reported in the figure legends. A Kruskal–Wallis test with Dunn's multiple comparisons was used to compare more than two groups. Correlation of individual data points was visualized by a linear regression fit and analyzed by computing Spearman r. For statistical evaluation of XY-plots, we used an RM two-way ANOVA test (repeated measurements/analysis) with Sidak's multiple comparisons. For the comparison of more than two groups in XY-plots, Tukey's multiple comparisons were applied. For the in-sample analysis of human cortical slices (paired experimental design), we used a Wilcoxon matched-pairs signed-rank test. $p$-values$<0.05$ were considered statistically significant (*$p<0.05$, **$p<0.01$, ***$p<0.001$); results that did not yield significant differences are designated 'ns'. Statistical differences in XY-plots were indicated in the legend of the figure panels (*) when detected through multiple comparisons, irrespective of their localization and the level of significance. In the text and figures, values represent the mean ± standard error of the mean (s.e.m.).

## Graphical illustrations

Figures were prepared using Photoshop graphics software (Adobe, San Jose, CA). Image brightness and contrast were adjusted.

## Acknowledgements

We thank Peter Jedlicka and Julia Muellerleile for helpful discussions. We thank Barbara Joch and Sigrun Nestel for their excellent technical assistance.

## Additional information

### Competing interests

Thomas Deller: TD received funding from Novartis for a lecture on human brain anatomy. The other authors declare that no competing interests exist.

### Funding

| Funder | Grant reference number | Author |
| --- | --- | --- |
| Faculty of Medicine | EQUIP-Medical Scientist | Maximilian Lenz |
| Faculty of Medicine | Berta-Ottenstein-Program | Jakob Straehle |
| Deutsche Forschungsgemeinschaft | CRC 1080 | Thomas Deller Andreas Vlachos |
| Deutsche Forschungsgemeinschaft | CRC 974 | Andreas Vlachos |

The funders had no role in study design, data collection and interpretation, or the decision to submit the work for publication.

### Author contributions

Maximilian Lenz, Conceptualization, Formal analysis, Funding acquisition, Investigation, Visualization, Methodology, Writing - original draft, Project administration, Writing - review and editing; Pia Kruse, Amelie Eichler, Formal analysis, Investigation; Jakob Straehle, Funding acquisition, Methodology, Project administration; Jürgen Beck, Resources, Methodology, Project administration, Writing - review and editing; Thomas Deller, Resources, Funding acquisition, Writing - original draft, Writing - review and editing; Andreas Vlachos, Conceptualization, Resources, Supervision, Funding acquisition, Visualization, Methodology, Writing - original draft, Project administration, Writing - review and editing

**Author ORCIDs**
Maximilian Lenz (iD) https://orcid.org/0000-0003-3147-4949
Pia Kruse (iD) https://orcid.org/0000-0002-1742-1608
Amelie Eichler (iD) https://orcid.org/0000-0001-7990-654X
Jakob Straehle (iD) https://orcid.org/0000-0003-3063-8972
Jürgen Beck (iD) https://orcid.org/0000-0002-7687-6098
Thomas Deller (iD) https://orcid.org/0000-0002-3931-2947
Andreas Vlachos (iD) https://orcid.org/0000-0002-2646-3770

## Ethics

Human subjects: Human brain tissue was obtained from a local biobank operated through the Department for Neurosurgery at the Faculty of Medicine, University of Freiburg (AZ 472/15_160880). All experiments carried out in this study were approved by the local ethics committee (AZ 593/19).

Animal experimentation: All experiments carried out in this study were performed according to the German animal welfare legislation and local authorities [approved by the animal welfare officers of the Faculty of Medicine at the University of Freiburg (AZ X-17/04C)]. Animals were kept in a 12 h light/dark cycle with access to food and water ad libitum. Every effort was made to minimize distress and pain of animals.

## Decision letter and Author response

Decision letter https://doi.org/10.7554/eLife.63026.sa1
Author response https://doi.org/10.7554/eLife.63026.sa2

# Additional files

## Supplementary files

• Supplementary file 1. Cortical resection samples.

• Transparent reporting form

## Data availability

All data generated or analysed during this study are included in the manuscript and supporting files. Source data files have been provided for all figures and supplementary material. Data and statistical analysis (Software: GraphPad Prism) are accessible through the following link (Dryad platform): https://doi.org/10.5061/dryad.6djh9w102.

The following dataset was generated:

| Author(s) | Year | Dataset title | Dataset URL | Database and Identifier |
|---|---|---|---|---|
| Lenz M, Kruse P, Eichler A, Straehle J, Beck J, Deller T, Vlachos A | 2021 | All-Trans Retinoic Acid induces synaptic plasticity in human cortical neurons | https://doi.org/10.5061/dryad.6djh9w102 | Dryad Digital Repository, 10.5061/dryad.6djh9w102 |

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
