## [Decision Letter]

**Acceptance summary:**

Your study reporting the acute effect of retinoic acid on excitatory synaptic transmission and its underlying mechanisms in human cortical neurons is very timely and important. As the function of retinoic acid in the brains of vertebrate animals beyond early development has not been extensively explored, your work will be of general interest to many readers in the field of neuroscience, especially those who work in the areas of synaptic plasticity.

**Decision letter after peer review:**

Thank you for submitting your article "All-Trans Retinoic Acid induces synaptic plasticity in human cortical neurons" for consideration by *eLife*. Your article has been reviewed by three peer reviewers, one of whom is a member of our Board of Reviewing Editors and the evaluation has been overseen by John Huguenard as the Senior Editor. The following individual involved in review of your submission has agreed to reveal their identity: Michael A Sutton (Reviewer #2).

The reviewers have discussed the reviews with one another and the Reviewing Editor has drafted this decision to help you prepare a revised submission.

All three reviewers are highly enthusiastic about the study reporting the acute effects of retinoic acid on excitatory synaptic transmission and its underlying mechanisms. The experiments are well executed and the results convincing. The reviewers agree that this work is of broad interest and suitable for publication at *eLife*. Aside from some minor comments that require minimal additional experiments or further clarification, the reviewers expressed one major concern regarding the dentate gyrus LTP data. Although further experiments are required to clarify the concerns, the reviewers recommended removing the LTP figure from the present study as it is not well connected with the rest of the study. Our expectation is that the authors will eventually carry out the additional experiments and report on how they affect the relevant conclusions either in a preprint on bioRxiv or medRxiv, or if appropriate, as a Research Advance in *eLife*, either of which would be linked to the original paper.

1) The LTP experiments are a bit problematic for several reasons. First, it was done in mouse hippocampal DG neurons, not in cortical neurons. The effect of RA may be different in different neuronal types, as has been shown in previous mouse studies. Without knowing the effect on basal transmission in these neurons, it is hard to interpret the LTP results. Second, why did WT DG show no LTP? Third, previous work by Arendt et al., (2015) showed that RA enhances hippocampal CA1 neuron basal EPSCs, and occludes further LTP. The observation here in the DG with RA treatment points the opposite direction. Can the authors offer some explanation? When was RA injected relative to the LTP experiments? Does RA induce metaplasticity in DG neurons? Again, knowing the effect of RA on basal transmission specifically in the DG neurons at the same time point of LTP experiment would be informative toward understanding the effect on LTP. Given the complexity of the LTP experiments and the focus of the study, which is on basal transmission, we suggest the authors to consider removing this data from the current study and perform a more thorough investigation in a follow-up study.

2) Regarding the experiments in synaptopodin deficient mice, in both WT neurons (stated in main text, not in figure) and rescue neurons (Figure 3B), the baseline sEPSC amplitudes are significantly smaller than those of the KO neurons. Some discussion of this observation is warranted.

3) Figure 1C illustrates responses of layer 2/3 pyramidal neurons to intracellular current injection. While the passive membrane properties are quantified and similar regardless of atRA exposure, it is not clear if atRA affects intrinsic excitability of these neurons (i.e., the number of spikes elicited by different levels of injected current). These data should be included.

4) The pharmacological treatments (ActD, anisomycin etc.) in this study are in general very long (6 hr) compared to conventional methods (less than 2 hr). To control for potential toxicity associated with prolonged treatment, vehicle control should be added in both Figure 5 and Figure 6.

---

## [Author Response]

All three reviewers are highly enthusiastic about the study reporting the acute effects of retinoic acid on excitatory synaptic transmission and its underlying mechanisms. The experiments are well executed and the results convincing. The reviewers agree that this work is of broad interest and suitable for publication at eLife. Aside from some minor comments that require minimal additional experiments or further clarification, the reviewers expressed one major concern regarding the dentate gyrus LTP data.1) The LTP experiments are a bit problematic for several reasons. First, it was done in mouse hippocampal DG neurons, not in cortical neurons. The effect of RA may be different in different neuronal types, as has been shown in previous mouse studies. Without knowing the effect on basal transmission in these neurons, it is hard to interpret the LTP results. Second, why did WT DG show no LTP? Third, previous work by Arendt et al., (2015) showed that RA enhances hippocampal CA1 neuron basal EPSCs, and occludes further LTP. The observation here in the DG with RA treatment points the opposite direction. Can the authors offer some explanation? When was RA injected relative to the LTP experiments? Does RA induce metaplasticity in DG neurons? Again, knowing the effect of RA on basal transmission specifically in the DG neurons at the same time point of LTP experiment would be informative toward understanding the effect on LTP. Given the complexity of the LTP experiments and the focus of the study, which is on basal transmission, we suggest the authors to consider removing this data from the current study and perform a more thorough investigation in a follow-up study.

We fully agree that the translation of findings from one brain region to another – and specifically between corresponding regions of the mouse and human cortex – is challenging. The effects of all-trans retinoic acid (atRA) on excitatory neurotransmission (i.e., the main focus of the present study) are readily observed in both mouse and human cortical slices. In contrast, synaptopodin-deficient superficial (layer 2/3) pyramidal neurons do not exhibit atRAmediated synaptic strengthening. While it is worth noting that atRA-mediated synaptic plasticity is not observed in the dentate gyrus of synaptopodin-deficient mice, the reviewers have raised several valid, important points here (e.g., pertaining to baseline synaptic transmission and possible metaplastic effects in the mouse dentate gyrus). Indeed, the work of Arendt et al., (2015) must be carefully considered in this context. Following the reviewers’ recommendation, we decided to remove these results from the revised version of our manuscript and will use them in a follow-up communication. Julia Muellerleile and Peter Jedlicka, who were responsible for performing these experiments, are no longer listed as coauthors. Instead, their input has been acknowledged in the revised manuscript. We will address the reviewers’ questions in a detailed follow-up study, “atRA-mediated (meta)plasticity in the mouse dentate gyrus”. However, our results obtained in the mouse dentate gyrus in vivo will remain available to the research community via the preprint of the present manuscript on bioRxiv (Lenz et al., 2020; bioRxiv, doi: https://doi.org/10.1101/2020.09.04.267104).

2) Regarding the experiments in synaptopodin deficient mice, in both WT neurons (stated in main text, not in figure) and rescue neurons (Figure 3B), the baseline sEPSC amplitudes are significantly smaller than those of the KO neurons. Some discussion of this observation is warranted.

We thank the reviewers for this careful observation. Previous work revealed no major differences in the baseline structural and functional properties of excitatory synapses in the dentate gyrus or CA1 between wild type and synaptopodin-deficient mice (e.g., Deller et al., 2003; Vlachos et al., 2013). However, based on the data in the present manuscript, layer 2/3 pyramidal neurons of the mouse mPFC appear to behave differently.

To address the reviewers’ concern, we carried out another series of single-cell recordings in cortical slices from synaptopodin-deficient mice and their age- and time-matched wild type littermates. While the basic intrinsic properties of these cortical neurons were not significantly different between the groups (new Figure 3—figure supplement 1), the amplitudes and frequencies of sEPSCs (spontaneous excitatory postsynaptic currents) were increased in synaptopodin-deficient preparations (new Figure 3—figure supplement 2). Transgenic expression of GFP/Synpo rescued the effects of synaptopodin deficiency on sEPSC amplitudes (but not frequency) and restored the ability of neurons to express atRA-mediated excitatory synaptic strengthening (c.f., revised Figure 3). Results obtained from wild type mice are now presented in Figure 3.

Accordingly, the text of the revised manuscript now reads:

“In Synpo^-/-^ preparations, no significant changes in sEPSC properties were observed following atRA treatment (Figure 3C, D), thus demonstrating the relevance of synaptopodin in atRA-mediated synaptic plasticity. Additionally, a reduction in the input resistance was not observed in atRA-treated Synpo^-/-^preparations (Figure 3—figure supplement 1). Furthermore, the active and passive membrane properties of Synpo^+/+^ and Synpo^-/-^ neurons were indistinguishable (Figure 3—figure supplement 1), despite significant increases in the baseline sEPSC properties of Synpo^-/-^ preparations (Figure 3—figure supplement 2).”

We provide further speculation and offer potential avenues for future research in the Discussion, as follows:

“Considering that increased sEPSC amplitudes and frequencies were observed in the cortical neurons of Synpo^-/-^ mice, it would be interesting to evaluate whether alterations in retinoid metabolism are present in this mouse model.”

“Similarly, the role of synaptopodin in atRA-mediated changes in intrinsic cellular properties, e.g., reductions in the input resistance of principal neurons in the mouse cortex, warrant further investigation, particularly in the context of recent work on the distinct electrical properties of human and murine cortical neurons (Gidon et al., 2020).”

3) Figure 1C illustrates responses of layer 2/3 pyramidal neurons to intracellular current injection. While the passive membrane properties are quantified and similar regardless of atRA exposure, it is not clear if atRA affects intrinsic excitability of these neurons (i.e., the number of spikes elicited by different levels of injected current). These data should be included.

Thank you for this comment. Action potential frequencies as a function of injected current are presented in the revised version of our manuscript, as alterations therein were observed at high current injections in the presence of actinomycin D or anisomycin (see revised Figure 1, Figure 3—figure supplement 1 and Figure 4—figure supplement 1). Interestingly, a reduction in the input resistance of layer 2/3 pyramidal neurons was seen in wild type mice but not in synaptopodin-deficient animals. However, since our study focused on the structural and functional changes of excitatory synapses in the human cortex, further follow up on these interesting findings is outside the scope of the current manuscript.

Accordingly, the text now reads: “Since no major changes in active or passive membrane properties were detected in these initial experiments, we focused on the effects of atRA on excitatory synapses and dendritic spines.”

Results pertaining to intrinsic cellular properties are provided in the supplementary information.

4) The pharmacological treatments (ActD, anisomycin etc.) in this study are in general very long (6 hr) compared to conventional methods (less than 2 hr). To control for potential toxicity associated with prolonged treatment, vehicle control should be added in both Figure 5 and Figure 6.

This is an important point that needs further clarification. Thank you. During the experiments with acute slice preparations, we did not observe any overt signs of toxicity. For example, we established a giga-ohm seal in whole-cell recordings of neurons in each group with equal success. The plasma membranes behaved normally when approached with the patch pipette. Likewise, no apparent signs of extensive cell death within the tissue were evident under brightfield microscopy. However, we decided to err on the side of caution and carry out the following additional experiments:

As suggested by the reviewers, vehicle-only controls were included in experiments with each pharmacological agent (i.e., actinomycin D and anisomycin); these experiments confirmed our primary results (new Figure 4). Please note that baseline synaptic transmission in the mouse neocortex was not affected by anisomycin or actinomycin D (Figure 4). Similarly, anisomycin did not affect sEPSC properties in human cortical slices (Figure 5). Although these experiments do not directly demonstrate a lack of toxicity, alterations in neurotransmission and network activity are readily observed when neurons die.

We also analyzed the active and passive membrane properties of all neurons recorded in Figure 4 (see Figure 5—figure supplement 1). Accordingly, the text now reads:

“To control for possible toxic effects, the basic active and passive membrane properties of cortical slices treated with actinomycin D or vehicle-only were recorded (Figure 4—figure supplement 1). Other than a reduction in AP frequencies at high-current injections, we did not observe any significant differences between the two groups (Figure 4—figure supplement 1). Moreover, baseline sEPSC properties were indistinguishable between actinomycin D- and vehicle-treated slices (Figure 4A). However, we observed a significant increase in the amplitudes of AMPA receptor-mediated sEPSCs in the atRA-treated group in both vehicle-only and actinomycin D co-incubated slices (Figure 4A, B), thus further confirming that atRA mediates synaptic strengthening.”

“Anisomycin incubation had no major effects on passive membrane properties, while reductions in AP frequencies were observed at high-current injections (Figure 4—figure supplement 1). In contrast to our experiments with actinomycin D (Figure 4C), however, no significant changes in sEPSC amplitudes were detected in the presence of both atRA and anisomycin (Figure 4C, D).”

Please also note that synaptopodin cluster sizes, as well as the sizes of synaptopodin(+) and synaptopodin(-) spine heads (Figure 5), were comparable in control and anisomycin-treated human cortical slices (c.f., Figure 2). Therefore, we are confident that our results cannot be readily attributed to neurotoxic effects of actinomycin D and anisomycin.